# Perspectives of people experiencing homelessness with recent non-fatal street drug overdose on the Pharmacist and Homeless Outreach Engagement and Non-medical Independent prescribing Rx (PHOENIx) intervention

**Natalia Farmer** [1]*, **Andrew McPherson**[2], **Jim Thomson**[3], **Richard Lowrie**[2,4]

1 School of Social Work, Glasgow Caledonian University, Glasgow, United Kingdom, 2 Pharmacy Services, NHS Greater Glasgow and Clyde, Glasgow, United Kingdom, 3 We See You Project, Glasgow, United Kingdom, 4 Centre for Homelessness and Inclusion Health, University of Edinburgh, Edinburgh, United Kingdom

* Natalia.farmer@gcu.ac.uk

**Data Availability Statement:** Data cannot be shared publicly because participants were

## Abstract

### Introduction

In Scotland, a third of all deaths of people experiencing homelessness (PExH) are street-drug-related, and less than half of their multiple physical- and mental health conditions are treated. New, holistic interventions are required to address these health inequalities. PHOENIx (Pharmacist Homeless Outreach Engagement and Non-medical Independent prescribing Rx) is delivered on outreach by National Health Service (NHS) pharmacist independent prescribers in partnership with third sector homelessness charity workers. We describe participant's perspectives of PHOENIx.

### Methods

This study aims to understand experiences of the PHOENIx intervention by participants recruited into the active arm of a pilot randomised controlled trial (RCT). Semi-structured in-person interviews explored participants' evaluation of the intervention. In this study, the four components (*coherence*, *cognitive participation*, *collective action*, *reflexive monitoring*) of the Normalisation Process Theory (NPT) framework underpinned data collection and analyses.

### Results

We identified four themes that were interpreted within the NPT framework that describe participant evaluation of the PHOENIx intervention: differentiating the intervention from usual care (*coherence*), embedding connection and consistency in practice (*cognitive participation*), implementation of practical and emotional operational work (*collective action*), and lack of power and a commitment to long-term support (*reflexive monitoring*). Participants

interviewed, with interviews transcribed verbatim. It would therefore be inappropriate for this to be made public given that participant identification (and possibly non-participants) together with sensitive data may inadvertently occur. This project (21/22/0004) was reviewed and approved by Health Research Authority, East of Scotland Research Ethics Committee 01, NHS Lothian, Waverly Gate, 2-4 Waterloo Place, Edinburgh EH1 3EG, United Kingdom. Any requests for sharing of data must be sent jointly to this committee, Sandra Wylie (sandra.wylie@nhslothian.scot.nhs.uk) and to last author Dr Richard Lowrie (richard.lowrie.ed. ac.uk). We may be able to share redacted data on a case-by-case basis, but only after seeking individual participant approval.

**Funding:** The authors received funding from NHS Greater Glasgow and Clyde and Scottish Government Drug Deaths Taskforce. The funders had no role in study design, data collection and analysis, decision to publish, or presentation of the manuscript.

**Competing interests:** The authors have declared that no competing interests exist

successfully engaged with the intervention. Facilitators for participant motivation included the relationship-based work created by the PHOENIx team. This included operational work to fulfil both the practical and emotional needs of participants. Barriers included concern regarding power imbalances within the sector, a lack of long-term support and the impact of the intervention concluding.

## Conclusions

Findings identify and describe participants' evaluations of the PHOENIx intervention. NPT is a theoretical framework facilitating understanding of experiences, highlighting both facilitators and barriers to sustained engagement and investment. Our findings inform future developments regarding a subsequent definitive RCT of PHOENIx, despite challenges brought about by challenging micro and macro-economic and political landscapes.

## Introduction

People experiencing homelessness (PExH) have multiple unmet health- and social care needs [1], contributing to premature mortality [2]. PHOENIx (Pharmacist Homeless Outreach Engagement and Non-medical Independent prescribing Rx), a collaborative and complex health- and social care intervention, aims to address these needs through weekly assertive outreach by National Health Service (NHS) pharmacists and third sector homelessness caseworkers. By providing intensive and holistic health- and social care support and persistent follow-up, a key objective is to reduce overdose and emergency service utilisation. Participants are visited via outreach and seen by the same pair of PHOENIx workers (pharmacist and third sector worker) at least once per week, with priorities set by the person experiencing homelessness and tackled gradually, in succession. As well as providing immediate medical care, health assessment and medication prescribing, PHOENIx offers practical support e.g. advocacy, support for application for welfare benefits, clothing, phones. Working collaboratively with key stakeholders, (in particular with Glasgow's Specialist Homelessness GP practice and Glasgow Royal Infirmary Emergency Department who collectively provided clinical governance for PHOENIx), community alcohol- and drug recovery services, and social care services, previous qualitative research has shown that PHOENIx may improve health [3]. Moreover, initial quantitative studies carried out on the PHOENIx intervention highlights that it can have a positive impact on the health-related quality of life of individuals [4].

The PHOENIx intervention is set against a backdrop of half of all deaths in Scotland in people experiencing homelessness being drug-related [5]. There is, however, a paucity of research on interventions to tackle the intersecting issues of homelessness and problem drug use [6], despite homelessness itself having a bi-directional causal pathway with illicit drug use and overdose [7]. Across Scotland, drug-related death (DRD) rates are increasing and remain one of the highest in the world. In Glasgow, over 70% of PExH have problem street-drug use [8] and, while most are receiving care from Alcohol and Drug Recovery Services (ADRS), including daily prescribed opiate replacement therapy, they continue to overdose. There is a need for interventions that prevent premature mortality and frequent emergency department visits by people experiencing homelessness [9]. People experiencing homelessness also have a higher incidence of post-traumatic stress disorder, major depression and other chronic health conditions, including blood borne viruses, contributing to increased readmission rates to hospital [10]. Permanent, supportive accommodation can impact positively on mental health and

wellbeing in people experiencing homelessness [11], but is only part solution with the main critical attribute of health-seeking behaviour of people experiencing homelessness being distrust in healthcare from stigma-associated behaviours by health providers [12]. PHOENIx aims to address DRDs in a subgroup of those at highest risk of DRD (homeless and recent non-fatal overdose) through outreach-based assessment and treatment of a wide range of health problems and health determinants, and interventions prioritised by the patient [1, 13].

There is an urgent need for evaluation of innovative approaches to address the public health crisis of drug-related deaths in Scotland. Definitive RCTs are the gold standard approach to demonstrating the efficacy of complex interventions and, prior to that, feasibility studies and pilot RCTs are recommended to confirm recruitment, retention, data collection, intervention fidelity, and barriers and facilitators to real world implementation [14]. A feasibility study of the PHOENIx intervention was conducted and suggested merit in progressing to a pilot RCT [4]. Subsequently, the PHOENIx after overdose randomised controlled trial (RCT) recruited 128 PExH with recent non-fatal street drug overdose, allocating 62 participants to PHOENIx intervention (weekly outreach visits by pharmacist and third sector worker in addition to care as usual, for 10 months), and 66 participants to care as usual [1, 13, 15]. Usual care in this study means health and social care that patients living in Glasgow receive as part of normal practice. This includes health and social care from specialist Alcohol and Drug Recovery Services (ADRS), from general practitioners, or from third sector homelessness charities. However, this health and social care is unlikely to be intense, joined-up, and patients are not likely to be assertively outreached or asked to prioritise their own needs, in order that they can be tackled systematically. A comprehensive examination of PHOENIx versus usual care from the pilot RCT has been published elsewhere [1].

Parallel process evaluations are becoming increasingly important in healthcare RCTs, because they can help to contextualise the delivery and receipt of public health interventions [16]. They are often recognised as an integral feature of a randomised controlled trial of a complex intervention [17] and can assist with interpretation of outcomes [18] due to inherent difficulties in defining or developing a new intervention [19]. Moreover, they help to provide insight into the "black box" of the intervention by investigating the discrete perceptions of participants [20]. Previous evaluations of PHOENIx have explored stakeholders' views, highlighting the ability of the intervention to provide a 'sticky', 'informal' and 'flexible' service that provides support to overcome barriers to health- and social care for hard-to-reach populations that fall through the gaps in care delivery [3]. In this study, evaluation focused on how the intervention was perceived by participants experiencing homelessness with a recent non-fatal overdose and in the context of a pilot RCT, as opposed to stakeholders delivering the intervention. This approach was chosen to investigate specific mechanisms of impact, such as participants' responses and interactions with PHOENIx [21], in order to facilitate careful consideration in regards to intervention improvement and development from a patient-led perspective. The objectives of this study are twofold;

- To understand potential barriers and facilitators to engagement with PHOENIx

- To differentiate between the PHOENIx intervention and usual care from a participant perspective

## Methods

### Design and setting

This qualitative study was part of a larger pilot randomised controlled trial (RCT) set in Glasgow, the largest city in Scotland and one of the largest in the UK. Recruitment to the pilot RCT

**Table 1. NPT construct in association with the implementation of PHOENIx.**

| NPT Construct | |
|---|---|
| Coherence | Sense-making work to understand the possibilities of an intervention. |
| | What is the purpose of PHOENIx? |
| Cognitive participation | Relational work that builds a community of practice around an intervention. |
| | What promotes participation with PHOENIx? |
| Collective action | Operational work that people enact to make an intervention function. |
| | How do participants interact with PHOENIx to make them work? |
| Reflexive monitoring | Appraisal work where people assess how a new practice affects them and others. |
| | How do participants appraise PHOENIx? |

started on the 11th of May 2021 and was completed on the 1st of September 2021. Informed written consent to participate in the embedded qualitative study was obtained by independent researcher AM as part of consent to participate in the RCT quantitative study.

We used Normalisation Process Theory (NPT) to understand and describe how participants in the active part of the RCT experienced and evaluated the PHOENIx intervention. NPT has roots in science and technology studies and can be seen as the antithesis of actor-network theory because of the assertion of explanation over description [22]. A systematic review of NPT for process evaluation of complex interventions found that it provides an important theoretical framework to support comprehension and implementation of interventions in transformational healthcare [23] and can be useful to understand the implementation of complex interventions in health care systems [24]. NPT offers a robust framework that focuses upon the processes and operational aspects of what individuals and teams do in order to implement an intervention [25]. This involves the use of four NPT constructs outlined in Table 1: *coherence* (sense-making work), *cognitive participation* (relational work), *collective action* (operational work), and *reflexive monitoring* (appraisal work). Together, these constructs help foster understanding in regards to how an intervention is implemented, operationalised and experienced within health- and social care settings. NPT has also been successfully utilised within qualitative research to further understand complex pharmacist-led interventions for medication safety in primary care [26] and general practice responses to opioid prescribing [27]. In this study, we aimed to use NPT to aid data collection and analyses in order to identify and understand how people experiencing homelessness with recent non-fatal overdose experienced and evaluated the PHOENIx intervention.

## Participants and recruitment

Eligible participants were aged 18 years old or older, defined by the European Typology of Homelessness and Housing Exclusion (ETHOS) as homeless (roofless, houseless, insecure accommodation) with at least one non-fatal street-drug overdose, due to illicit drug use in the 6 months prior to recruitment to the study. The inclusion criteria were;

Homeless (living in temporary homeless accommodation, no fixed abode, or rough sleeping);

and

At least one self-reported, non-prescribed drug overdose (blackout/lack of response and slow/irregular breathing that was thought to progress to complete cessation of respiratory effort unless treated) in the past 6 months confirmed by:

- A witness; or

- An ambulance call out; or

- An emergency department (ED) visit; or

- An administration of naloxone.

Exclusion criteria were; living in residential or community-based rehabilitation facility that had direct access to in-house medical and nursing care; or unable to provide written informed consent.

The PHOENIx intervention is a complex health and social care intervention, with generalist pharmacists with an independent prescribing qualification, paired with a third sector homelessness outreach worker. Patients would be seen by the pharmacist and third sector outreach worker at least once per week, with priorities set by the patient. Results of the PHOENIx pilot RCT are described elsewhere [1]. Participants not randomised to the intervention arm of the study received care as usual.

Twenty participants were purposively sampled by independent researchers AM and NF from the intervention arm of the PHOENIx pilot RCT [13], using a heterogeneous sampling technique in order to ensure maximum variation and to capture a range of perspectives across gender, age, accommodation and number of prior overdoses. No individuals declined participation in this study, therefore the participant response rate was 100%. We believe that the high response rate is a result of the psychotherapeutic and trusting relationship between participants and members of the PHOENIx team. Interviews were carried out in private in third sector drop-in venues for people experiencing homelessness, or in homeless shelters. Privacy was maintained by conducting interviews in a private space and where both the interviewer and participant felt safe and comfortable. These are dedicated spaces in third sector organisations, especially designed for counselling sessions and other types of intimate one-to-one conversations. In order to maintain confidentiality and anonymity, participants were asked to provide a pseudonym for dissemination of findings. Participant characteristics are outlined in Table 2.

## Data collection

Twenty face-to-face semi-structured interviews were conducted by NF, an academic qualified social worker with experience in qualitative research with marginalised individuals and communities and not involved in the delivery of the trial intervention. Interviews were conducted between November 2021 and January 2022 in-person in emergency- or hostel accommodation or in a private space within a homeless charity (Simon Community Scotland) drop-in hub. Interview questions were guided by the NPT framework and lasted a median duration of 40 minutes, with a range from 30 to 60 minutes. The interview template was designed with feedback from individuals from a third sector charity with lived- and living experience of homelessness and overdose. This was to ensure rigour in the formulation of research questions by identifying gaps or problematic areas, such as appropriate language/terminology, and revised accordingly. Following interviews, each participant was given a pseudonym to ensure anonymity and confidentiality. Participants received a £10 shopping voucher in recognition for their participation in the interviews. Interviews were recorded using an encrypted electronic device and transcribed verbatim using an external organisation approved by the NHS.

## Data analysis

Normalisation Process Theory (NPT) was utilised in order to gain a deeper understanding of participants' evaluations of the PHOENIx intervention. An initial reflexive thematic analysis (RTA) [28] was conducted by NF and AM after transcripts were entered into NVivo Version 12 software. This approach was taken in order to mitigate against simply 'shoehorning' [29] data into the predetermined NPT constructs. This enabled orientation with the data,

**Table 2. Participant characteristics (N% or mean (SD) or median (range).**

| | |
|---|---|
| **Sex** | |
| Male | 13 (65%) |
| Female | 7 (35%) |
| **Age** | |
| Mean (SD) | 45 (8.7) |
| **Number of overdoses (past 6 months)** | |
| Median (range) | 2 (1–30) |
| **Accommodation** | |
| Supported | 6 (30%) |
| Unsupported | 12 (60%) |
| Hospital* | 2 (10%) |
| **Opiate Substitute Therapy** | |
| Methadone | 18 (90%) |
| Buprenorphine | 1 (5%) |
| Nil | 1 (5%) |
| **Methadone dose (mg)** | |
| Mean (SD) | 90 (22.8) |
| **Received diazepam prescription** | |
| Yes | 1 (5%) |
| No | 19 (95%) |
| **Naloxone** | |
| Possess Naloxone | 12 (60%) |
| Knows how to use it | 17 (85%) |
| **Alcohol and Drug Recovery Service (ADRS)** | |
| Generic ADRS | 13 (65%) |
| Homeless ADRS | 6 (30%) |
| Not known to ADRS | 1 (5%) |

*Although hospital was part of the exclusion criteria to the study, these participants were admitted to hospital post-recruitment

generation of initial codes, developing themes and reviewing them with the team (NF, AM, RL and JT), alongside consideration of researcher positionality within knowledge production. Following this stage, data themes were mapped to a coding framework informed by the NPT constructs: *coherence* (sense-making work); *cognitive participation* (relational work); *collective action* (operational work); and *reflexive monitoring* (appraisal work). In order to ensure rigour and reliability during the process of data analysis, NF and AM adopted the reflexive pair method [30], to consider researcher positionality within knowledge production and ensure consistency in the generation of codes and developing themes. Additionally, the involvement of co-authors (RL and JT) provided additional insight and feedback that helped to develop iterative revisions and ensure further consistency and credibility during the interpretation and written analysis.

## Patient and public involvement

People experiencing homelessness were involved at each stage of the pilot RCT, including initially identifying a need for the study and recruitment to it. People with lived and living experience were asked for their views on the materials used, such as consent forms, participant information sheets and baseline assessment tools. As part of the trial steering group, we met

weekly with people with lived and living experience of homelessness and drug-related over-dose [1]. Regarding the qualitative component of the study, we trialled the questionnaire schedule on a person with lived experience of homelessness and drug-related problems and adjusted it accordingly.

## Ethical approval and governance

The study was approved by South East Scotland National Health Research Ethics Committee 01. REC reference 21/SS/0004. All participants were provided with participant information sheets and were able to ask questions about the study and use of data before providing informed written consent. Consent from participants was sought at the beginning of the Pilot RCT and then clarified prior to the interviews. All participants consented to the interviews being recorded. All data were collected and stored in accordance with NHS General Practice Data for Planning and Research (GPDPR) guidelines. Trial registration number: ISRCTN10585019.

## Results

We used the consolidated criteria for reporting qualitative research (COREQ) as a framework to guide reporting of our research. Four components promoted further understanding into how participants evaluated the implementation of PHOENIx into practice and was informed by the NPT constructs of *coherence* (what is the purpose of PHOENIx), *cognitive participation* (what promotes participation with PHOENIx), *collective action* (how do participants interact with PHOENIx) and *reflexive monitoring* (how do participants appraise and rate their care provided by PHOENIx). The four components of NPT and how the participants evaluated PHOENIx alongside this are outlined in Table 3. In this section we explore the perceptions of participants across the components to consider how they evaluated the intervention. Specific attention is given to our aim of understanding factors that either promoted or inhibited the incorporation of PHOENIx in their everyday lives. In turn, identifying any gaps or issues that would require the intervention to be resigned or improved in future delivery.

## Coherence: Differentiating the PHOENIx intervention from usual care

The first NPT construct, *coherence* refers to the sense-making work that participants went through and shared in interviews regarding their understanding of the purpose and possibilities of the PHOENIx intervention. Specifically, participants revealed an important element of this process rested upon their perceptions of the management of drug overdose in general and the roles of other different health- and social care providers. Within the data, contrasting perceptions of their usual care in comparison to the PHOENIx intervention were consistently expressed by participants. Provision provided by the PHOENIx team was differentiated from other health- and social care teams, with participants stating their unhappiness with Alcohol and Drugs Recovery Services (ADRS) in contrast to their satisfaction with the intervention. Many described their usual care as inadequate as opposed to PHOENIx, which was understood to be beneficial because it provided aspects of care and support lacking from other health- and social care providers.

**Table 3. Overview of themes identified through the NPT constructs.**

| Coherence | Cognitive Participation | Collective Action | Reflexive Monitoring |
|---|---|---|---|
| Differentiating the intervention from usual care | Embedding connection and consistency in practice | Implementation of practical and emotional operational work | Lack of power and a commitment to long-term support |

For example, participants expressed negative opinions in regards to the realities of ADRS management and significant frustration in terms of their ability to successfully manage their drug use and health problems. There was an overall sense of being 'dismissed' when visiting their addictions teams alongside a distinct lack of help:

> You go in [laughs] they see you, "How are you, how's your drug use been this week, are you feeling all right, how's your injecting site?" They don't look at you or anything like that, they just ask you questions. "Right, do you think you need up 10ml or another 20ml on your methadone? No, right, here's script for 2 weeks or whatever." That's not help. (Sean, male, age 53, unsupported hotel accommodation, Glasgow city)

> I mean everything is coming from myself, they've not helped, I mean I can't say that they have because they've not, and I'm not going to sit here and say that they've done something that they haven't. (Clare, female, age 46, unsupported hotel accommodation, Glasgow city)

On the other hand, participants significantly differentiated PHOENIx from the usual care they received and, overall, participants provided positive feedback of PHOENIx and in terms of the management of drug- and health problems, the purpose of the intervention was perceived as providing a service that was often missing from other teams. Hence, the benefits of the intervention were clear, as the below two quotes highlight:

> I found [PHOENIx staff] more supportive than my drug worker, you know what I mean? She [ADRS worker] was increasing me instead of decreasing me [Methadone]. Then she wasn't doing the call she should be doing, calling me to see if everything was okay. How you doing with your methadone? She wasn't doing anything like that. It was [PHOENIx staff] who was contacting me. (Daniel, male, age 47, supported hostel, Glasgow city)

> The [ADRS team] are piss, but what they [PHOENIx staff] do is brilliant, you know, it's like night and day with the [ADRS team] and they [PHOENIx staff] they [PHOENIx staff] says, "I'm going to do something." and they do it. (Brian, male, age 48, supported accommodation, Glasgow south)

Overall, the intervention was perceived as beneficial, because it was seen to be providing essential support regarding the management of drug overdose in areas of health- and social care that participants stated was absent.

## Cognitive participation: Embedding connection and consistency in practice

*Cognitive participation* refers to the relational work that is required and involves engaging and motivating individuals around a new intervention. A component of this construct is 'enrolment', which entails the complex work that is involved in getting individuals to 'buy in' to new practices or processes. It influences both motivation and commitment, and is an important aspect in regards to ensuring successful participation. The findings revealed that participants highly valued both the (1) connection and (2) consistency that members of the PHOENIx team provided in order to build and sustain a relationship and actively engage individuals with the intervention. In conjunction, these two aspects created the motivation that was required for individuals to both initially engage with a new service and to continue their sustained involvement with the team on an ongoing basis.

Participants frequently highlighted the relational connection that they had with members of the PHOENIx team, stating that the genuine and caring approach taken enabled them to feel willing and motivated to spend time with the team. The non-judgemental approach shown

by PHOENIx staff was vital in order to create trust and confidence with participants, placing an emphasis on authenticity and sincerity. Consequently, this related to participants feeling able to trust the team, which in turn created feelings of comfort and care, as highlighted by the below quotes:

> I've been brought up in care and I've had a really hard life so I can see through a bullshitter, I can see through somebody who genuinely cares or are just doing their job. I can tell if they actually give a shit or not. (Anna, female, age 37, hotel accommodation, Glasgow city)

> I've got a good relationship with them. Very supportive, and they care, they really do care. I trust them. And it is that trust. They've been there for me, they've been supportive, and that's what I needed or else I think I'd have done something daft a long time ago. (Sarah, female, age 37, supported accommodation, Glasgow south)

> Comfortable. Every time we were with them, they made us feel comfortable. They didn't speak down to me. I just felt we were on an even keel with them, do you know what I mean? Aye, just being comfortable and you can just tell when somebody is on your side. (Caroline, female, age 43, hotel accommodation in Glasgow city)

Secondly, consistency was emphasised as an integral aspect that enabled participants to sustain continued participation and motivation during the trial intervention. Importantly, this was related to the level of meaningful connection that has been established with members of the team and their ability to create long-lasting relationships that were dependable. The following are examples that highlight the important of consistency and reliability within relational work that helped to empower and sustain connections.

> They were the only workers really consistent. My care manager, my other worker, I don't know her as well as I know [PHOENIx team], if you know what I mean. Any time I was in trouble I would go to one of them. I can just tell that they care. (Anna, female, age 37, hotel accommodation, Glasgow city)

> Just seeing [PHOENIx team] on a weekly basis is beneficial for me [. . .] That's a goal for me, seeing people like that motivates me to stay sober. It's embarrassing when you have to see them on the Wednesday and you took something you shouldn't have, you're like pure wounded, pure defeated. That feeling of being defeated is powerful and I don't want that in my life anymore. (Brian, male, age 48, supported accommodation, Glasgow south)

> I think they're brilliant because you're not going to get penalised by missing an appointment [. . .] they're not going to be on your back, they're not going to walk out on you. A couple of times I got really depressed, thinking, I'm not letting them in, I don't want to see anybody and I thought, they're not going to want to see us again and they came out. (Caroline, female, age 43, hotel accommodation, Glasgow city)

In general, participants described the connection that they developed with members of the PHOENIx team as an essential component required for their participation with the intervention. Given the extensive stigma and discrimination encountered by those who use drugs and are homelessness, for the intervention to be successful, this particular aspect is crucial in terms of both initial 'buy-in' and continued investment and retention. Therefore, establishing that the personal attitudes and skill set of the PHOENIx team in their ability to build an authentic relationship with participants is an essential aspect of the intervention in terms of securing trust and confidence.

## Collective action: Implementation of practical and emotional operational work

*Relational integration* is a component of the *collective action* construct that refers to the knowledge work enacted by individuals that enables confidence and accountability within a new intervention. This aspect of PHOENIx was emphasised significantly and discussed in detail during the interviews. It provided clarity in regards to the work that the PHOENIx team operationalised that was most meaningful and useful from the perspectives of participants. The findings focused upon two sub-themes that participants identified as most valuable for them in terms of managing the ongoing stress burdens they encountered in their everyday lives struggling to meet their health- and social needs. Specifically, this referred to the interactional and integrated outreach work that the pharmacists and third sector caseworkers conducted in fulfilling both the (1) practical needs and (2) emotional needs of participants.

The practical outreach the team provided was a key factor that the majority of participants highlighted as most useful and highly valued, because it reduced the individual stress burdens experienced by the cohort. This was especially crucial given the known discrimination and barriers encountered by PExH when attempting to access health- and social care within the context of homelessness and drug use. The knowledge base and advocacy skills of the PHOENIx team in regards to addressing immediate healthcare needs during outreach visits was reported by all participants. This included the ability of pharmacists to treat some health conditions e.g. wound care, assess severity of health conditions e.g. respiratory problems, and prescribe for all health conditions except problem drug use (Alcohol and Drug Recovery Services in Glasgow stipulated PHOENIx pharmacists were not to prescribe for problem drug use) on outreach, as described within the below quotes:

> They got me an appointment with the hospital, they've rearranged appointments for me at the [acute hospital] because I missed that, all because I've been taking seizures for the last couple of years, a couple of them are drug-induced seizures and the [PHOENIx team] rearranged the appointment for me because I forgot about it, because I didn't have it written down. (Brian, male, age 48, supported accommodation, Glasgow south)

> They helped my legs. I've had eight abscesses in my legs now. I've had all sorts of blood clots. If they didn't get me the help I was getting at the time, I'd probably would have lost that one. They found out I had a heart murmur. I think they recognised that I've got a lot of internal stuff as well I'm having to deal with. (Lee, male, age 48, hotel accommodation, Glasgow west)

Participants also stressed the importance they gained from the intervention in regards to the advocacy that the team provided with social care, which in turn alleviated the stress burden associated with navigating various social welfare systems. Again, the knowledge and skills of third sector staff to provide support with Universal Credit, Personal Independence Payments (PIP) and housing applications was frequently reported, and these systems without support elevated participants anxiety and stress.

> We never had Wi-Fi, never had a computer, never had a phone. Sometimes, things were building up on my phone, things we had to do, it was just all building up and [PHOENIx team] would come out and just ask us what we needed done and he would sit there for ages on the computer doing the electricity thing and I've got [PHOENIx team] over here doing a PIP form for me, and it's taking him like 45 minutes just to get through and then he's on

the phone for an hour. Practical and I appreciated it. (Caroline, female, age 43, hotel accommodation, Glasgow city)

They [PHOENIx team] would address each new thing. They can open doors no other service can. They got me into a place when I was really desperate, cold, absolutely ill, you know what I mean? They were always trying to get me accommodation. Somewhere safe to stay, you know? (Lee, male, age 48, hotel accommodation, Glasgow west)

In addition, the emotional support that participants reported they received throughout the intervention was consistently highlighted as a vital aspect that served to reduce stress levels by providing hope and humanity within the context of a highly stigmatised environment. The relational element of outreach visits emerged as an important quality and participants described an increase in confidence with each regular visit, helping them to maintain their motivation, with something to look forward to.

They come in and help and stuff, and just having somebody to talk to, they keep my spirits up, do you know what I mean? And then that plants that seed in my head, not to touch drugs. Every morning I wake up I'm like, right, no drugs today. (Sarah, female, age 37, supported accommodation, Glasgow south)

They gave me a wee bit of hope, do you know what I mean? Not easy at first, but they made me feel comfortable and didn't make me feel belittled or anything, you know? The way they speak to you, they treat you like a human being rather than see you like a junkie. So, it gives you hope, do you know what I mean? (Lee, male, age 48, hotel accommodation, Glasgow west)

It's actually took the stress off me. I feel less stressful. I feel a lot better. (Daniel, male age 47, supported hostel, Glasgow city)

Together, the combined practical and emotional qualities embedded within PHOENIx outreach visits fostered a level of confidence within participants that mitigated against the increased stress burdens many encountered in their attempts to navigate health- and social care systems. This enabled individuals to receive immediate, accessible and potentially life-saving healthcare and social advocacy that they would otherwise be unable to access.

## Reflexive monitoring: Lack of power and a commitment to long-term support

*Reflexive monitoring*, the appraisal work that is crucial to understand how the PHOENIx intervention was assessed was limited at the individual level, with many participants struggling to retain engagement towards the end of the interview and, notably, repetition in answers was evident. However, when asked if they would recommend the PHOENIx intervention, the overall response was highly positive and all had or would recommend to others due to the characteristics of the intervention explored in this paper. While no specific negative issues arose, there were changes and anxieties that may lead to the reconfiguration of the intervention, or, at least, an opportunity to redefine or modify communication with participants at the outset.

An initial aspect that participants suggested would have been beneficial during the intervention involved the incorporation of regular food parcels during outreach visits. Additionally, some participants were keen to see members of the PHOENIx team have more 'power' and felt that PHOENIx staff regularly had their 'hands tied' and were unable to enact as much change as desired in various areas, such as mental health provision:

The [PHOENIx] workers to get a wee bit more say in things. That's about it. If they did get more of a say in things, they could do a wee bit more for addicts. I think you should be given that wee bit more power of things. If you were given that wee bit more power there would be a lot more people's lives saved. (Lee, male, age 48, hotel accommodation, Glasgow west)

They tried to get me some counselling, and they got knocked back. They got me listed for the [third sector organisation], so I'm listed for counselling, but there's no date yet, but I know I need counselling. (Brian, male, age 48, supported accommodation, Glasgow south)

Furthermore, it is important to recognise that towards the end of the interview, participants strongly expressed their anxiety that the PHOENIx intervention and outreach visits were not ongoing. Consequently, this seemed to have a negative impact on individuals, with many aware that once visits and the trial ended, the commitment to long-term support would end. As the below highlights:

Because my journey is just starting and I am scared I will relapse. I would like to come out of the treatment centre and still have their support, because it's still going to be fresh. I'm scared in case I relapse and nobody is there to help me. (Anna, female, age 37, hotel accommodation, Glasgow city)

I feel as if I'm going to lose contact with [PHOENIx team]. And eventually not see them again. That'll just knock me right back into depression again. (Matthew, male, age 41, hotel accommodation, Glasgow city)

## Discussion

The purpose of this study was to gain an understanding into how participants experiencing homelessness with a recent non-fatal overdose evaluated PHOENIx in order to identify factors that either promoted or inhibited the incorporation of the intervention into their everyday lives. Informed by NPT, analysis revealed several factors within the four constructs that participants emphasised as critical to the success of the intervention, alongside gaps in implementation that could be enhanced or redesigned for future delivery.

The sense-making work (*coherence*) needed to understand the possibilities of an intervention was an important aspect in participants understanding of the purpose of PHOENIx. As previous research has demonstrated, the intervention is highly effective in terms of engaging with people experiencing homelessness who encounter barriers when attempting to access health- and social care, leading to many falling through the gaps [3]. However, these findings provide further understanding in regards to why initial engagement with PHOENIx is so successful in the management of drug overdose. Participants overwhelmingly reported that the intervention was crucial for them due to a consistent sense of being failed within other health- and social care services [31]. There was significant differentiation between the intervention and the usual care that they received, creating a clear purpose and benefit in PHOENIx. The management of drug overdose from other service providers, particularly ADRS teams was described negatively, leading to the importance of PHOENIx in providing essential support that many stated was lacking elsewhere.

For the intervention implementation to be successfully sustainable, a crucial aspect that participants emphasised was the relational work the PHOENIx team effectively created (*cognitive participation*). This was related to how engaged and motivated participants felt working with the team during the trial period. Guidance suggests the importance of 'buy-in' with a new intervention [32]. However, within a context marred by previous relational injury such as stigma [33], this study underscored the critical necessity for strong and supportive

relationships, whereby non-judgemental characteristics within the team are central to creating and maintaining participation. This was evidenced in terms of the ability of the PHOENIx staff to create what participants described as a genuine, authentic and sincere approach. Furthermore, participants emphasised that two specific components, 'connection' and 'consistency' were key in the success of building relationships with the team, which, in turn, fostered trust, confidence and continued reliability.

PHOENIx is provided through a combination of NHS employee pharmacists and third sector (Simon Community Scotland and The Marie Trust) individuals handpicked with the requisite human relational skills and non-judgemental approach to care. In terms of how participants experienced the intervention, this collaborative skill-set was crucial and highlighted to be the most useful aspect of PHOENIx for several reasons, due to the implementation of both practical and emotional operational work (*collective action*). Practically, participants described a reduction in their daily stress burdens, as the team were able to immediately treat some health conditions and organise or rearrange health- and social care appointments. Previous research has highlighted the importance measuring treatment burden for clinical trials [34]. Our findings support this and provide more detail in the context of homelessness and drug overdose, and also aligns well with emerging signs of delayed overdoses, emergency department visits and hospitalisations in the intervention arm of the pilot randomised controlled trial, but these returned to pre-PHOENIx rates after the intervention visits ended [1]. Moreover, participants stressed that challenges navigating various social welfare systems, in particular Universal Credit, PIP and housing applications created additional burdens that increased levels of anxiety. Advocacy and knowledge that third sector members of the team provided were highly valued, which is most relevant given the increasing extent of digital exclusion among people living in areas of socioeconomic disadvantage and the ubiquitous role that digital technologies play in health- and social care systems [35]. Participants further acknowledged that the emotional support provided by the team not only provided much needed hope within the stigmatised context of homelessness and drug use [33], but it also increased their motivation and sustained engagement and retention within the intervention. Given that research has gaps within service delivery in regards to mental health and counselling within the context of addiction service [36], it is worth noting how stronger mental health provision might be incorporated into the future intervention design.

Overall, participants evaluated the PHOENIx intervention highly (*reflexive monitoring*), with all participants stating that they would recommend the intervention and expressed a desire for outreach visits to continue for reasons highlighted in this discussion. However, there was a sense among participants that the trial was due to end, creating anxiety and disappointment. Recent guidance has highlighted the importance of supportive, caring and therapeutic relationships [37], yet in the context of the closure of the specialist GP service for PExH in Glasgow [38], and the associated closure of PHOENIx outreach, the ability to commit to long-term support is questioned. Furthermore, when asked what participants would like to see change, evaluations pointed towards structural issues within service delivery, as many reported that they would like to see the PHOENIx team have extended permissions to manage mental health and problem drug use on outreach.

It is important to note that this study is not set in a vacuum, but within dynamic political and structural environments. Financial reductions to homelessness services and increasing numbers of homelessness presentations purports that there is an ever greater need for services to address and adapt to the consequential health- and social care shortfall. Finally, study findings are particularly apposite given the recent decision by the local health- and social care partnership to discontinue the specialist homelessness GP service and the PHOENIx service in Glasgow, despite available evidence suggesting merit in retaining both.

## Conclusions

This research has utilised NPT to reveal important factors for successful implementation of the PHOENIx intervention from the perspectives of participants. It is important to acknowledge that this study has limitations, in that it explored participant experiences of the intervention and did not include wider stakeholders. Previous studies on PHOENIx have brokered stakeholder observations and while these studies did not look at PHOENIx in the context of a pilot RCT, they did nevertheless describe the intervention. Unlike the current study, previous qualitative work did not involve participants who were, by design, at higher risk of street-drug overdose. In this study, our aim was to focus on participant evaluations in order to inform changes to the intervention in use, and in turn, prevent overdoses and worsening health. Additional limitations involve potential selection bias of participants due to the involvement of Simon Community Scotland staff in this study. In order to mitigate against this, independent researchers AM and NF conducted the sampling process and a consensus approach by the research team was implemented to ensure a wide range of experience across participant characteristics. While this included variety in regards to gender and age, it is important to acknowledge the study did not capture an ethnically diverse sample. Under-representation of racially minoritised individuals in regards to their engagement with addiction services in Scotland is recognised [39] and future research is essential to examine this knowledge gap in order to inform service delivery for those who are racially marginalised at the intersection of homelessness and drug use.

Given our key aim, a significant strength within this study involved the inclusion of those with lived- and living experience of homelessness and non-fatal overdose within the methodological design and analysis of data. The robust approach towards inclusion of lived experience, alongside patient and public involvement, described earlier in this paper, ensured that feedback was sought at each phase of the research process to ensure rigour and trustworthiness. Consequently, this developed our understanding of what factors are needed to ensure engagement and continued retention from the perspectives of participants. Powerful components of the intervention were the connection and consistency fostered by the PHOENIx team in order to create both initial 'buy-in' and continued engagement. Findings suggest this rests on the relational skill set of the team in order to create authentic and meaningful relationships with participants. Practical and emotional operational work provided by both pharmacists and third sector caseworkers was significantly valued in order to reduce stress levels of participants and foster much needed hope within a highly stigmatised context. Aspects that could optimise support include the incorporation of mental health provision, expanded responsibility for prescribing for problem drug use, and food parcels within the delivery of the intervention.

We have devised a number of recommendations for future research from this study. These are:

- Based on the outcomes of this research and results from the associated pilot RCT, we recommend that PHOENIx is further investigated in a definitive multicentre RCT;

- Further qualitative investigation of the PHOENIx intervention be undertaken in a multicentre RCT to gauge the intervention in different geographical settings, and over a longer period of time;

- Finally, we recommend Normalisation Process Theory (NPT) as a method, not only to help explain complex interventions, but also as a framework to help explain the social dynamics inherent with participation in research.

## Supporting information

**S1 File. Logic model: PHOENIx intervention for people experiencing homelessness with a recent non-fatal overdose.**
(DOC)

**S2 File. Interview guide.**
(DOC)

## Author Contributions

**Conceptualization:** Natalia Farmer, Richard Lowrie.

**Data curation:** Andrew McPherson, Jim Thomson, Richard Lowrie.

**Formal analysis:** Natalia Farmer, Andrew McPherson, Jim Thomson, Richard Lowrie.

**Methodology:** Natalia Farmer.

**Writing – original draft:** Natalia Farmer, Andrew McPherson, Jim Thomson, Richard Lowrie.

**Writing – review & editing:** Natalia Farmer.

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
