## [Decision Letter · Decision Letter 0]

12 Mar 2024

PONE-D-24-01041Perspectives of people experiencing homelessness with recent non-fatal street drug overdose, on the Pharmacist and Homeless Outreach Engagement and Non-medical Independent prescribing Rx (PHOENIx) interventionPLOS ONE

Dear Dr. Farmer,

Thank you for submitting your manuscript to PLOS ONE. After careful consideration, we feel that it has merit but does not fully meet PLOS ONE’s publication criteria as it currently stands. Therefore, we invite you to submit a revised version of the manuscript that addresses the points raised during the review process.

We look forward to receiving your revised manuscript.

Kind regards,

Andrew Scheibe, MBChB MPH

Academic Editor

PLOS ONE

“The authors received funding from NHS Greater Glasgow and Clyde and Scottish Government Drug Deaths Taskforce.”

Additional Editor Comments:

Thank you for submitting this paper.

Please review and respond to the reviewer comments.

Additional things to address:

In methods, describe the ratio of people approached/ invited to participate and any people that declined participation (response rate).

Table 2: methadone dose - please correct to show what the dose was in mg (it should not be mg/ml).

The description of limitations seems to be limited to "participant experiences of the intervention." Consider reflecting on potential selection and information bias that may have influenced the findings and how these where mitigated and any potential influence they may have had on the findings.

The conclusion could be strengthened. Consider including additional recommendations around future research, and and on the utility of using the NPT framework in research of this nature and in other contexts.

Reviewers' comments:

Reviewer's Responses to Questions

**Comments to the Author**

1. Is the manuscript technically sound, and do the data support the conclusions?

Reviewer #1: Yes

Reviewer #2: Yes

2. Has the statistical analysis been performed appropriately and rigorously? 

Reviewer #1: N/A

Reviewer #2: N/A

3. Have the authors made all data underlying the findings in their manuscript fully available?

Reviewer #1: Yes

Reviewer #2: No

4. Is the manuscript presented in an intelligible fashion and written in standard English?

Reviewer #1: Yes

Reviewer #2: Yes

5. Review Comments to the Author

Reviewer #1: Thank you for the opportunity to review your manuscript. It was most interesting to read, and one has a clear sense that this is crucial advocacy work regarding services to people who use drugs and who are experiencing homelessness. I really do hope that the publication of your work will have a positive impact on the restoration of the person-centred services you describe for these marginalised groups in your area.

The manuscript is very well-written, clear and easy to read and follow. The settings is well-described, and the methodology is sound. The presentation of results is clear, and the discussion is integrated. The conclusion gives a good summary of the work.

I have made some minor editing amendments via track changes. There are a few comments for your attention as well.

In particular:

Check the reference style, as it seems inconsistent. Use Vancouver and check consistency throughout. I have edited most of the references, but please double check. See the PLoS ONE referencing guidelines: https://journals.plos.org/plosone/s/submission-guidelines#loc-references

Reviewer #2: Technical soundness:

1. The aim of the trial is explained in some sections, but the objectives are missing; need to be stated.

2. The study population and the two study arms are not clear, briefly describe.

3. Define “Usual care” that the other group was on.

4. Ethical considerations: 4.1 Clarify where the interviews were done and how privacy was ensured.

4.2 There are names attached to quotes from participants, clarify what measures were taken to ensure anonymity and confidentiality of the data – lines 255, 258, 268, …….

5. Clarify the sentence that starts on line 529: our robust approach ensured that feedback was sought at each phase of the research process. Describe the phases and from whom feedback was sought.

6. On the interview guide, clarify why the question on line 59 is only about pharmacists.

7. There is no consent form attached, so clarity is needed on whether participant consent was sought for recording the interviews.

Data availability: The response to the requirement is "Yes - all data are fully available without restriction". However, it needs to be made clear whether this availability is in line with the requirement of " The data should be provided as part of the manuscript or its supporting information, or deposited to a public repository"

6. PLOS authors have the option to publish the peer review history of their article (what does this mean?). If published, this will include your full peer review and any attached files.

Reviewer #1: **Yes: **Michelle NS Janse van Rensburg

Reviewer #2: No

---

## [Author Response · Author response to Decision Letter 0]

10 Apr 2024

Dear Drs Scheibe and Janse van Rensburg 

All authors associated with this research would like to thank you for your professional insight and your recommended amendments to our manuscript, which enhance it and provide additional clarity for the reader. We are in a state of flux in Glasgow with a reduction in drug and alcohol funding from central and devolved governments. The service that we provide to these patients is crucial, yet the powers that control matters have decided to terminate it, alongside specialist GPs working in homeless health. We would like to point you to our recent findings publication https://bmjpublichealth.bmj.com/content/2/1/e000219 which paints a very stark picture of addictions, services for people experiencing homelessness and drug-related deaths. Despite this, there is hope that, publications like this one, will reach the decision-makers and make them realise the dedication, commitment, psychotherapeutic skills and compassion workers in PHOENIx have and that this works for patients, and just maybe the service can see a reversal of fortunes, and we will see PHOENIx rise from the ashes and on the streets of Glasgow once more.

Academic Editor

1. We have checked PLOS ONE’s style requirements and amended as required (author contributions have been explored in more detail, the abstract word count now sits at 300 words, stated patient and public involvement in the study and updated the references, including the reference list.

2. Financial statement should now read;

The authors’ received funding from NHS Greater Glasgow and Clyde and the Scottish Government Drug Deaths Task Force. The funders had no role in study design, data collection and analysis, decision to publish, or presentation of the manuscript.

3. Unsure of the data sharing protocols, we wrongly provided details of data sharing for the quantitative part of our Pilot RCT. Data sharing of interviews wold not be appropriate because of the (very real) potential for identification of participants and associated sensitive data. The data availability statement should now read;

The full manuscripts from the semi-structured interviews are not available as they contain potentially identifying and sensitive participant information. Participants were not informed or requested to allow transcripts to be shared publicly. 

4. We have reviewed our reference list for completion and correctness. Furthermore, we have updated it to provide further contextual evidence.

Additional Editor Comments:

Methods and ratio of people approached invited to participate – all people approached/invited to participate in this study. We have stated this and provided a potential rationale for the high response rate. 

Table 2. Methadone dose. This has now been change to mg only.

The limitations have been expanded to include potential bias from researchers and possible mitigation, with potential influence on study findings.

We have strengthened the conclusion by providing recommendations for future research and the utility of NPT as a research method.

Reviewer 2

We have checked the referencing with the Vancouver referencing guide and amended it accordingly. We wish to thank Dr Janse van Rensburg with her assistance in this matter.

1. The objectives of the study are now clearly stated with bullet points.

2. The inclusion and exclusion criteria for the study has been added and a brief description of the intervention and usual care has been given with a reference for more detail on these

3. Usual care has been defined within the text and readers are directed to the findings paper of the Pilot RCT for further details.

4.1 We have clarified where interviews were carried out and set out how we maintained privacy.

4.2 We asked participants to provide us with a pseudonym for the qualitative interviews in order to maintain privacy and confidentiality. We have stated this process in the manuscript text.

5. We have clarified the sentence; Our robust approach ensured feedback, and we have included a section on patient public involvement

6. With regards to the interview guide Line 59 mentioning only pharmacists, this should read “pharmacist and third-sector charity worker” and we have amended

7. We have clarified the issue of consent and detailed this in the manuscript

We have clarified the issue over data availability. Given that participants were interviewed and this was transcribed, it would be inappropriate for this to be public given that participant identification together with sensitive data may inadvertently occur.

---

## [Editor Report · Decision Letter 1]

17 Apr 2024

Perspectives of people experiencing homelessness with recent non-fatal street drug overdose on the Pharmacist and Homeless Outreach Engagement and Non-medical Independent prescribing Rx (PHOENIx) intervention

PONE-D-24-01041R1

Dear Dr. Farmer,

We’re pleased to inform you that your manuscript has been judged scientifically suitable for publication and will be formally accepted for publication once it meets all outstanding technical requirements.

Kind regards,

Andrew Scheibe, MBChB MPH

Academic Editor

PLOS ONE
---

## [Editor Report · Acceptance letter]

1 May 2024

PONE-D-24-01041R1 

PLOS ONE

Dear Dr. Farmer, 

I'm pleased to inform you that your manuscript has been deemed suitable for publication in PLOS ONE. Congratulations! Your manuscript is now being handed over to our production team.

Kind regards, 

on behalf of

Dr. Andrew Scheibe 

Academic Editor

PLOS ONE